# Genomic Characterization of Dengue Virus Outbreak in 2022 from Pakistan

**DOI:** 10.3390/vaccines11010163

**Published:** 2023-01-11

**Authors:** Massab Umair, Syed Adnan Haider, Zaira Rehman, Zunera Jamal, Qasim Ali, Rabia Hakim, Shaheen Bibi, Aamer Ikram, Muhammad Salman

**Affiliations:** Department of Virology, National Institute of Health, Islamabad 45500, Pakistan

**Keywords:** DENV-1, DENV-2, coinfection, whole genome sequencing, serotype, cosmopolitan genotype, dengue outbreak in Pakistan

## Abstract

Pakistan, a dengue-endemic country, has encountered several outbreaks during the past decade. The current study aimed to explore the serotype and genomic diversity of dengue virus responsible for the 2022 outbreak in Pakistan. From August to October 2022, NS-1 positive blood samples (n = 343) were collected from dengue patients, among which, (85%; n = 293) were positive based on RT-PCR. In terms of gender and age, dengue infection was more prevalent in male patients (63%; n = 184), with more adults (21–30 years; n = 94) being infected. The serotyping results revealed DENV-2 to be the most predominant serotype (62%; n = 183), followed by DENV-1 (37%; n = 109) and DENV-3 (0.32%; n = 1). Moreover, a total of 10 samples (DENV-2; n = 8, DENV-1; n = 2) were subjected to whole-genome sequencing. Among these, four were collected in early 2022, and six were collected between August and October 2022. Phylogenetic analysis of DENV-2 sequenced samples (n = 8) revealed a monophyletic clade of cosmopolitan genotype IVA, which is closely related to sequences from China and Singapore 2018, and DENV-1 samples (n = 2) show genotype III, which is closely related to Pakistan isolates from 2019. We also reported the first whole genome sequence of a coinfection case (DENV1-DENV2) in Pakistan detected through a meta-genome approach. Thus, dengue virus dynamics reported in the current study warrant large-scale genomic surveillance to better respond to future outbreaks.

## 1. Introduction

Dengue virus (DENV) is one of the most important arboviruses (arthropod-borne virus) in terms of public health concern and is known to cause dengue infection, primarily transmitted by the *Aedes aegypti* mosquito [1]. The dengue virus is an enveloped, spherical virus with icosahedral symmetry encapsulating 11-kilo bases long single-stranded positive-sense RNA [2] that encodes for one open reading frame having three structural and seven non-structural proteins [3]. Based on antigenic differences, four serotypes of dengue virus with 65% genome similarity have been identified, namely, DENV-1, DENV-2, DENV-3, and DENV4, besides a newly discovered DENV-5 in Malaysia, which usually circulates among non-human primates [4,5]. Within each serotype, there are multiple genotypes [6] that can be phylogenetically defined based on sequence variation in envelope (E) gene: DENV-1 (I-VI); DENV-2 (Asian I, Asian II, Asian/American, American, Cosmopolitan, and sylvatic); DENV-3 (I-V); and DENV-4 (Asian I, Asian II, Asian/American, American, Cosmopolitan, and sylvatic) [7,8]. These serotypes may induce a varied immunogenic effect by infecting different target cells, thus producing a high cytokine response, which in turn affects the disease severity. Moreover, secondary infection with a heterologous serotype may elicit a more prompt immune response than primary infection due to the antibody-dependent enhancement (ADE) mechanism [5]. Up to 40–80% of all dengue infections are asymptomatic, however, serotypes can induce the common symptoms of dengue fever with few eliciting the severe forms of dengue called dengue shock syndrome (DSS) and dengue hemorrhagic fever (DHF) [9,10]. The typical symptoms of dengue fever include pain in bones and muscles, headaches, abdominal discomfort, vomiting, and nausea. Symptoms of DHF include bleeding gums, fever, anemia, bloody intestinal discharge, and plasma leakage, causing respiratory failure. Additional signs of DSS include vascular system impairment, injury, and low blood pressure [11].

According to an estimate, every year more than 100 million dengue cases (more than 80% of the cases are mild/asymptomatic) are reported worldwide, with the highest number of infections from Southeast Asia [12]. As of 27 October 2022, dengue cases (n = 3,333,251) and deaths (n = 2984) had been reported globally [13]. Dengue fever was first recorded in Pakistan in 1994, according to the World Health Organization (WHO), and since then, the number of laboratory-confirmed cases had increased [14]. The country is now endemic for dengue, with several outbreaks reported in 2010, 2017, 2019, 2020, and 2021 [15,16]. Sporadic cases of dengue are observed throughout the year; however, the peak incidence is observed during the monsoon season (July–September) [17,18]. During 2022, the dengue outbreak was observed in the post-monsoon season (August–October, 2022), with Punjab, Khyber Pakhtunkhwa, and Sindh being the most affected provinces [14].

The co-circulation of multiple dengue serotypes increases the risk of co-infection with the frequency ranging from low (5–30%) to high (40–50%) [19,20]. In Pakistan, the presence of all four dengue serotypes is common, and the co-circulation of multiple DENV serotypes are reported along with co-infection cases in the last few years [21]. During the 2011 dengue outbreak in Pakistan, a high frequency of dengue co-infection (27%) with DENV-1, DENV-2, and DENV-3 was reported, and (13.8%) was reported for DENV-2 and DENV-3 co-infection [22,23]. Previous studies reported that serotype shift and co-circulation of multiple serotypes [24] can promote the incidence of dengue infections, thus, serotype identification necessitates surveillance to prevent a future outbreak. Therefore, the goal of the current study was to investigate the circulating serotypes during the 2022 outbreak in Pakistan and to explore the genomic diversity of the dengue virus by whole-genome sequencing. The study had reported the first coinfection of the whole genome case of DENV-1–DENV-2 in Pakistan using the metagenome approach.

## 2. Materials and Methods

### 2.1. Ethical Statement

The study was approved by the Internal Review Board of the National Institute of Health (NIH), Islamabad. Informed patient consent was obtained before sample collection.

### 2.2. Sample Collection

A total of 343 NS1-positive human serum samples of dengue collected between August and October 2022 were received at the Department of Virology, National Institutes of Health, Islamabad. Samples were collected along with the required information regarding age, gender, location, date of collection, and relevant clinical data.

### 2.3. RNA Extraction and Serotype Specific MultiPlex PCR Amplification

Using the QIAamp viral RNA mini kit (Qiagen, Hilden, Germany) according to manufacturer instructions, the viral RNA of all the samples was extracted. Serotype-specific multiplex, real-time reverse transcriptase PCR (rRT-PCR) was performed using CDC protocol [25]. A total of 25 μL multiplex reaction mixture was run for each DENV serotype (DENV-1 to DENV-4) on ABI-7500 real-time thermocycler using the Invitrogen one-step qRT-PCR kit (Thermo Fisher Scientific, Waltham, MA, USA).

### 2.4. Next Generation Sequencing

A subset (n = 18) of dengue-positive samples having Ct value <30 were randomly selected for whole-genome sequencing, among which ten samples were successfully sequenced. Whole-genome sequences of four samples collected in April 2022 and six from the September to October 2022 outbreak were generated. Briefly, this was performed using the metagenome library preparation kit, NEBNext^®^ Ultra II Directional RNA Library Prep Kit (New England Biolabs, Ipswich, MA, USA), and the total extracted RNA was processed with RNase-free DNase (Roche, Basel, Switzerland) and prepared for unbiased paired-end sequencing (2 × 150 bp) per the manufacturer’s instructions. By using the Qubit dsDNA HS test kit (Invitrogen, Waltham, MA, USA), the sequencing libraries were measured on the Qubit 4.0 fluorometer and then pooled appropriately in equimolar concentrations. The DNA 1000 Kit and Agilent Bioanalyzer (Agilent Technologies, Santa Clara, CA, USA) were utilized to measure the size of the libraries. Finally, at the Department of Virology, National Institute of Health, Islamabad, Pakistan, the pooled libraries were subjected to sequencing on the Illumina MiSeq platform using sequencing reagent, MiSeq Reagent Kit v2 (300-cycles) (Illumina, San Diego, CA, USA).

### 2.5. NGS Data Analysis

The resulting raw NGS reads were evaluated using the FastQC v0.11.9 program [26]. The Trimmomatic tool (v0.39) was used to remove low-quality base calls and adapter sequences using the parameters ILLUMINACLIP:adapters-PE.fa:2:30:10 LEADING:3 TRAILING:3 SLIDING WINDOW: 4:30 MINLEN:50 [27]. The PICARD tool (Picard MarkDuplicates) was used to remove PCR duplicates from the filtered reads [28], which were then assembled into contigs with the SPAdes program v3.15.5 using the default parameters [29] and compared to the NCBI Non-redundant (NR) database to map the reads with the most related genome using Burrows-Wheeler Aligner (BWA) [30]. Finally, using the Geneious Prime software v.2022.2 (threshold = 0%, Assign Quality = total, minimum coverage >10), the consensus genomes were called [31]. The sequences have been submitted to GenBank with accession numbers OP811977-OP811980, OP811982-OP811984, and OP898557-OP898559.

### 2.6. Phylogenetic Analysis

For phylogenetic analysis, the study sequences were subjected to a BLAST search to identify the closely related sequences. Based on BLAST search, the globally reported sequences of the dengue virus along with the reference genomes of all DENV genotypes were downloaded from NCBI (https://www.ncbi.nlm.nih.gov/, accessed on 14 November, 2022). The multiple sequence alignment was performed by MAFFT [32], and a scaled phylogenetic tree was generated by maximum likelihood (ML) using IQ-Tree v2.2.0 [33] software using GTR + G as a substitution model with a bootstrap confidence limit based on 1000 replicates. The substitution model was chosen using MEGA 11. The tree was visualized and annotated using FigTree v1.4.4 [34].

## 3. Results

During the study period the Department of Virology, NIH received serum samples of 343 dengue NS1-confirmed patients (serologically tested) from 11 different districts of the country. Serotyping was successful for 293 (85%) samples using RT-PCR. Among the PCR-positive cases, males were more commonly infected (n = 184; 63%) as compared to females (n = 109; 37%). The mean age of dengue-positive patients was 30.75 (±12.4) years (males = 29 (±12.2) years and females = 33 (±12.3) years. The most common presenting symptom was fever (98%), followed by myalgia (94%) and backache (70%). Hematological data were available for 144 patients, showing thrombocytopenia (platelet count ≤ 150,000) in 129 (89%) cases. The hematocrit level was less than 40% in 97 (68%) patients, between 41–45% in 32 (22%), and greater than 45% in 15 (10%) patients (Table 1).

Dengue serotyping results showed the dominance of DENV-2 (n = 183; 62%), followed by DENV-1 (n = 109; 37%). A single case of DENV-3 was also detected during the current study. Week-wise distribution of serotypes showed the dominance of DENV-2 from August till the third week of October; however, DENV-1 became the more prevalent serotype in the last week of October (Figure 1). The highest number of DENV-2 and DENV-1 cases were reported from Rawalpindi (n = 78; n = 57), followed by Islamabad (n = 50; n = 35) and Peshawar (n = 35; n = 08), respectively. The only case of DENV-3 was reported from Peshawar (Figure 1 and Figure 2).

Phylogenetic analysis based on the current study of whole-genome sequences of DENV-2 (n = 08) revealed the circulation of cosmopolitan genotype, whereas the two DENV-1 viruses were classified as genotype III. The cosmopolitan genotype of DENV-2 is further divided into IVA (1 and 2) and IVB (1 and 2). All the studied DENV-2 isolates were clustered within a single monophyletic clade of genotype IVA that is closely related to viruses from China and Singapore and different from the clade containing DENV-2 viruses reported from Pakistan (IVB) during 2008 to 2013. For phylogenetic analysis, four of the DENV-2 viruses (NIH-PAK-01/2022, NIH-PAK-02/2022, NIH-PAK-04/2022, and NIH-PAK-05/2022) collected during April 2022 from Kech district (Balochistan province) were sequenced and compared with the four isolates (NIH-PAK-07/2022, NIH-PAK-08/2022, NIH-PAK-09/2022 and NIH-PAK-12/2022) collected during the outbreak of September to October, 2022 from Rawalpindi, Islamabad, and Karachi. All these eight Pakistani DENV-2 sequences shared high nucleotide (97.1%) and amino acid (97.5%) homologies. The two DENV-1 isolates shared high identities (99.1–99.7%) and were clustered with viruses previously reported in Pakistan in 2019 and China, Bangladesh, and Singapore in 2016. Pakistani DENV1 isolates share 99.1–99.7% identities with the 2019 isolates of Pakistan and 2016 isolates of China and Singapore both at the nucleotide and amino acid levels. Using the meta-genome approach, we were able to detect a case of coinfection of DENV-1 and DENV-2 in a sample (NIH-PAK-11/2022) collected from Rawalpindi in April (Figure 3).

To elucidate viral amino acid variations, DENV-2 amino acid sequences were aligned, which revealed 99–100% identity, indicating that protein sequences of all DENV-2 strains were highly conserved. In the coding region, we found no nucleotide insertion or deletion. When compared with the reference strain (strain: 16681, Genbank accession number: NC_001474), 36 amino acid substitutions were identified in Pakistani DENV-2 viruses and 13 previously reported isolates from Pakistan (Figure 4A). Of these, six amino acid mutations occurred in the structural protein (prM/M and E), fourteen amino acid mutations occurred in the non-structural proteins, NS1–NS3, and sixteen amino acid mutations occurred in the non-structural proteins, NS4–NS5. The NS1 and NS5 genes were found to harbor a large number of amino acid substitutions compared to other viral proteins. When comparing the mutation profile of DENV-2 study sequences with DENV-2 cosmopolitan reference strain (strain: 98900663 DHF DV-2, NCBI accession number: BAD42415.1), we have identified 18 amino acid substitutions in the study sequences. Of note, these changes were present in envelope (M596V, V588I), NS1 (S855T, V952T/I, F953S, K1047R), NS2B (L1366F, V1404I), NS3 (K1490R), NS4A (I2182V), NS4B (A2262T), and NS5 (I2762V, K2878E, H3047Y, K3061R, A3139V, S3167N, D3317E). The important polar–nonpolar amino acid substitutions were observed as T892A (NS1), S903L (NS1), T2487A (NS48), and R2921 (NS5), whereas the nonpolar-polar amino acid substitutions were observed as V952T (NS1), F953S (NS1), and A2262T (NS4B). Other amino acid substitutions were either from polar–polar or from non-polar to non-polar amino acids. Comparatively, the amino acid composition of eight DENV-2 isolates shared similarities with DENV-2 strains that caused epidemics in Singapore and China. However, it bore significant variation from the DENV2 strain that caused previous outbreaks of dengue during 2008 to 2013. This variation may be due to differences in the sub-genotypes of cosmopolitan, as the current study sequences belong to the IVA1 clade, while previous sequences belong to the IVB1 and 2 clade.

The amino acid sequence alignment of study DENV-1 revealed that these two viruses shared a similar mutation profile. No nucleotide deletions or insertions were observed within the coding region. The comparison of all the study sequences with the reference strain (AAW64436) has revealed 26 amino acid substitutions in the structural and non-structural regions (Figure 4B). Of these, seven mutations were in the structural proteins (capsid and envelope), ten mutations were in the NS1-NS3 region, and eight mutations were in the NS4-NS5 region. While comparing with the previous isolates (ON908217, ON873938) from Pakistan, we have observed only five amino acid substitutions in the envelope (L748M), NS2A (N1190S), NS4A(P2196L) and NS5 (H2928N and V3135A) region. Envelope and NS1 were found to be having a large number of mutations. The prM/M is found to be highly conserved. Polar–nonpolar substitutions are T257A (Envelope), and nonpolar–polar substitutions are I903T (NS1).

## 4. Discussion

Dengue is a fast-emerging arboviral infection in many countries worldwide, including Pakistan. The prevalence of dengue infection has increased significantly in recent years, and more than half of the world’s population is at risk [14]. According to WHO, an estimated 100–400 million infections occur each year, and Southeast Asia contributes significantly to the overall disease burden [35,36]. The first confirmed outbreak of dengue from Pakistan was reported in 1994, but it was not until November 2005 that a yearly epidemic pattern began in Karachi [37]. Since then, Pakistan experienced sporadic dengue fever outbreaks, which eventually became significant epidemics in 2011, 2017, 2018, and 2019 [15]. Alarmingly, a significant increase in the number of cases was recorded after 2018 (n = 3204), i.e., 47,120 in 2019 and 52,000 in 2021 [16,38]. According to data from the National Institute of Health, Pakistan, during 2022 (January to November), Pakistan reported 75,450 cases of dengue, mostly from Khyber Pakhtunkhwa (n = 22,617), followed by Sindh (n = 22,174), Punjab (n = 18,626), Islamabad (n = 5384), Balochistan (n = 5205), and Azad Jammu and Kashmir (n = 1444). Moreover, 136 deaths had been reported (Sindh = 61; Punjab = 45; KP = 18; Islamabad = 11 and Balochistan = 1). A major reason for such an increase in dengue cases during 2022 has been heavy rainfalls in Pakistan, which lead to catastrophic floods causing widespread displacement of people and providing breeding sites for mosquitoes [39].

We investigated the serotype and genomic diversity of the dengue virus circulating in Pakistan during August to October 2022, when most of the cases were reported from the country. By analyzing the overall serotype diversity, the current study reported DENV-2 (62%) to be the most prevalent serotype, followed by DENV-1 and DENV-3. This is in agreement with the prevalence of dengue serotypes reported from Pakistan in 2019 [15,40], however, it is in contrast to study findings investigating the prevalence of serotypes from KP province in 2017. In 2017, DENV-2 (38%; n = 196) was reported as the dominant serotype, whereas DENV-3 (37%; n = 192) was co-circulating at high numbers. Moreover, during 2017, DENV-4 (11%; n = 56) was prevalent along with low circulation of DENV-1 (5%; n = 25) [41]. During dengue outbreaks from 2011 to 2013, all four serotypes were reported, with the highest frequency of DENV-2 and DENV-3 reported in Punjab and KP provinces [42]. Timely identification of DENV serotypes can aid in the triaging and managing of patients, enabling rapid clinical response and appropriate epidemiological surveillance of diseases during outbreaks [43]. One of the major challenges in dengue serotyping in Pakistan is the lack of laboratory facilities having such capacities. At present, the National Institute of Health serves as the central laboratory receiving samples from different parts of the country. To deal with future outbreaks, it will be important to strengthen provincial public health reference labs to do real-time surveillance of dengue serotypes across the country.

The current study reported a single case of DENV-3 from Peshawar in 2022, which was prevalent in the early 1990s in Pakistan and lasted until 2016. In Pakistan, all four DENV serotypes co-circulate in different periods, which can increase the likelihood of coinfection [20,44]. There was a severe dengue outbreak reported with 100% DENV-2 along with 5% samples having coinfection with DENV-3 from Pakistan in 2017 [45]. Of note, the current study also contributes by whole genome sequencing of a coinfection case “DENV-1 and DENV-2” in 2022 for the first time in Pakistan.

The phylogenetic analysis of eight whole-genome samples of DENV-2 showed a cosmopolitan genotype (subclade IVA1). These study isolates share 99% amino-sequence identity with viruses from China and Singapore, detected in 2018, which could be attributed to active virus exchange between these countries, as both countries are one of the most connected global aviation hubs, and trade expansion resonates with numerous inbound and outbound dengue virus dispersal links [46]. Before 2022, there was a lack of reported DENV genome sequences for four years in Pakistan. Former research in 2017 reported the presence of a DENV-2 cosmopolitan genotype IV (lineage A1) via Sanger sequencing, which bifurcated into two sublineages (IVA, IVB) also sharing high similarity with Singapore and China isolates. During that period (2017), lineage IVA1 was prevalent in the dengue outbreak in Peshawar city and was reported to be dominant in East Asian countries based on phylogenetic analysis [45]. It is important to highlight that the research isolates were distinct from the clade of previously reported isolates in 2008 and 2013 (IVB) from Pakistan [45,47], which may be due to in situ evolution or a result of an invasive strain that has replaced the former genotype. However, further investigation is required (Figure 3). In line with this, the significance of lineage replacement events was also well established previously [48]. As a supporting notion, prior research (2008–2013) on DENV-2 dynamics in Pakistan via partial and complete sequences also revealed two distinct clades (IVA, IVB) of cosmopolitan genotype IV (lineage B2). During that period (2008 to 2013), the introduction and dissemination of lineage IVB2 were more successful in Pakistan than IVB1 [49], which is contrary to the current study trend. A previous study conducted a neutrality and selection pressure test, explaining that the positive selection regions (capsid, envelope, and non-structural proteins; NS2A NS3) of lineages A1 and B2 had different evolutionary processes [46]. They propose using this technique on the entire genome to shed further light on the evolution of dengue lineages in Pakistan, wherein the current study can play a role.

The current study phylogenetic analysis reported the genotype III of DENV-1 two whole-genomes samples sharing a similar clade with previous 2019 Pakistan isolates (ON908217.1 and ON873938.1). This sharing of a similar clade represents the common ancestry of study isolates with viruses detected in 2019. However, the study conducted in 2011 in Pakistan reported DENV-1 exhibiting genotypes 1 and IV via partial sequencing [49], which differed from the study-reported genotype. Our study isolates also share similarities with China and Singapore. In China, Genotype III was known to cause invasion and replacement of genotype II from 2013 to 2014 [50]. Whereas, the clade replacement is linked with genotype 1 of DENV-1 [51], which was previously found in 2011 in Pakistan [49]. Overall, the study found no major differences based on sequence similarity at amino acid (DENV-2; 97.5%, DENV-1; 99.1–99.7%) levels between the sequenced samples collected from the two time periods (April, Sep to Oct) in 2022 in Pakistan.

Notably, the unique mutations found in the envelope (E) gene of the DENV-2 coinfected sample were “Q332H and V588I”, and, in the NS5 gene, “R2589K, I2762V, K2878E, R2921G, H3047Y, K3061R, G3098V, Q3135K, G3096V, Q3135K, and A3139V” mutations were reported. Further, in DENV-1, we have reported an “L748M” mutation in the envelope region and “H2928N, V3135A” in the NS5 gene of the coinfected patient. The E and NS5 genes are responsible for infectivity, antigenicity [52,53], genome replication, host protein interactions, and drug target, respectively [54,55]. In particular, the whole genome sequence under study lacks two amino acid mutations (S70A; pre-membrane region, N610G; C-terminal domain of NS5) required for RdRp-activity in flaviviruses, which, during 2008 to 2013 in Pakistan, were reported as sites under positive selection pressure (*p*-value < 0.08) in the full genome [47]. While DENV-2 has the NS5-K861R mutation, which is unique to Pakistan strains between 2006 till 2022, its role remains unknown [56]. In light of the increasing rate of dengue infections throughout the world despite vector-control measures, several dengue vaccines had been developed and are currently undergoing different phases of clinical trials [57]. The only licensed dengue vaccine (CYD-TDV, Dengvaxia; Sanofi Pasteur, Lyon, France) is advised for use in individuals ≥9 years who have a history of dengue infection [58]. Therefore, for seronegative individuals, Qdenga, a tetravalent dengue vaccine candidate (TAK-003), was developed by Takeda, which is designed to protect against all the serotypes [59,60,61,62]. The Qdenga’s FDA approval process is going ahead along with granting a priority review [63]. Furthermore, this vaccine is all set to be launched, in Indonesia, early next year [64,65]. In mid-October 2022, the European Medicines Agency’s Human Medicines Committee (CHMP) has also recommended the use of Qdenga in Europe [66]. In Pakistan, most existing interventions relying on mosquito control are ineffective at preventing disease in the country. It is one of the most affected countries and majorly contributes to the global burden of dengue. Hence, Pakistan urgently needs to consider the use of a vaccine given that the current study has reported a diverse range of dengue serotypes and genotypes in 2022.

## 5. Conclusions

Altogether, the current study findings reported the prevalence of two major serotypes, DENV-2 (cosmopolitan genotype) and DENV-1 (III genotype), co-circulating in diverse geographic locations of Pakistan. The incidence of mixed infection with multiple serotypes and genotypes could increase the severity of the disease during future outbreaks in Pakistan. Thus, the current study’s findings on dengue viral dynamics call for large-scale genomic surveillance to better respond to future outbreaks in Pakistan.

## Figures and Tables

**Figure 1 vaccines-11-00163-f001:**
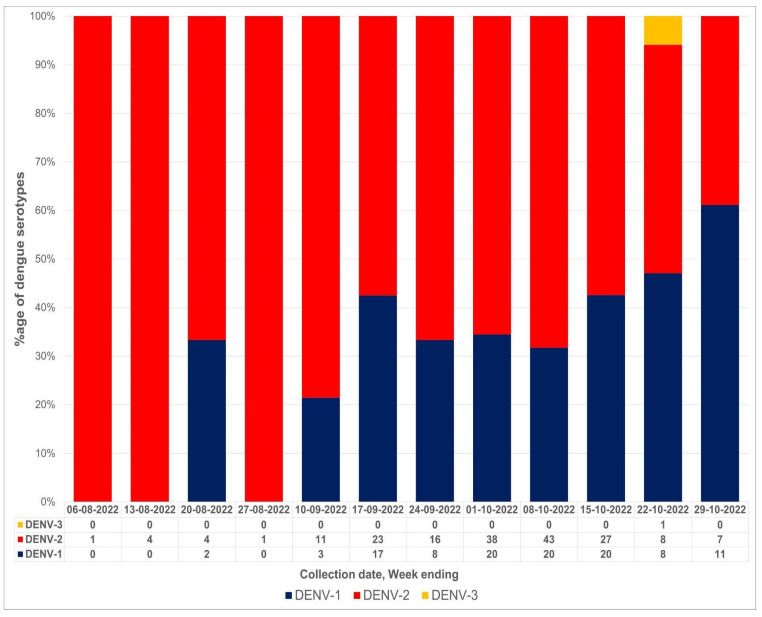
Week-wise distribution of dengue virus serotypes (August–October 2022).

**Figure 2 vaccines-11-00163-f002:**
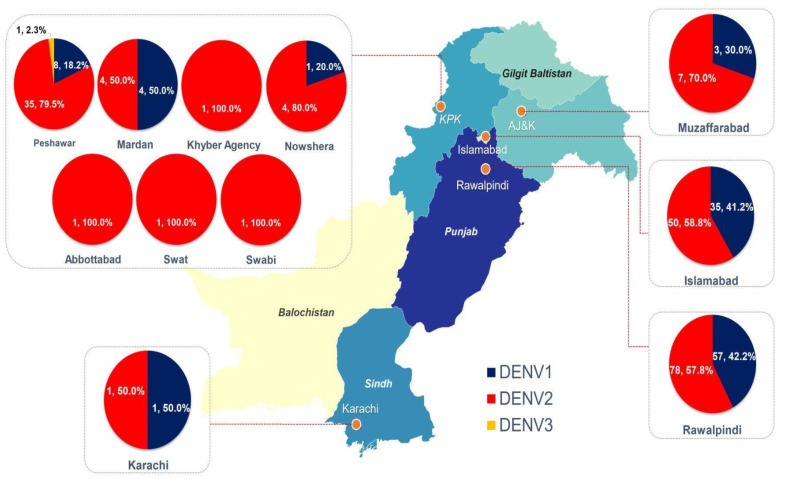
Geographical distribution of circulating dengue serotypes.

**Figure 3 vaccines-11-00163-f003:**
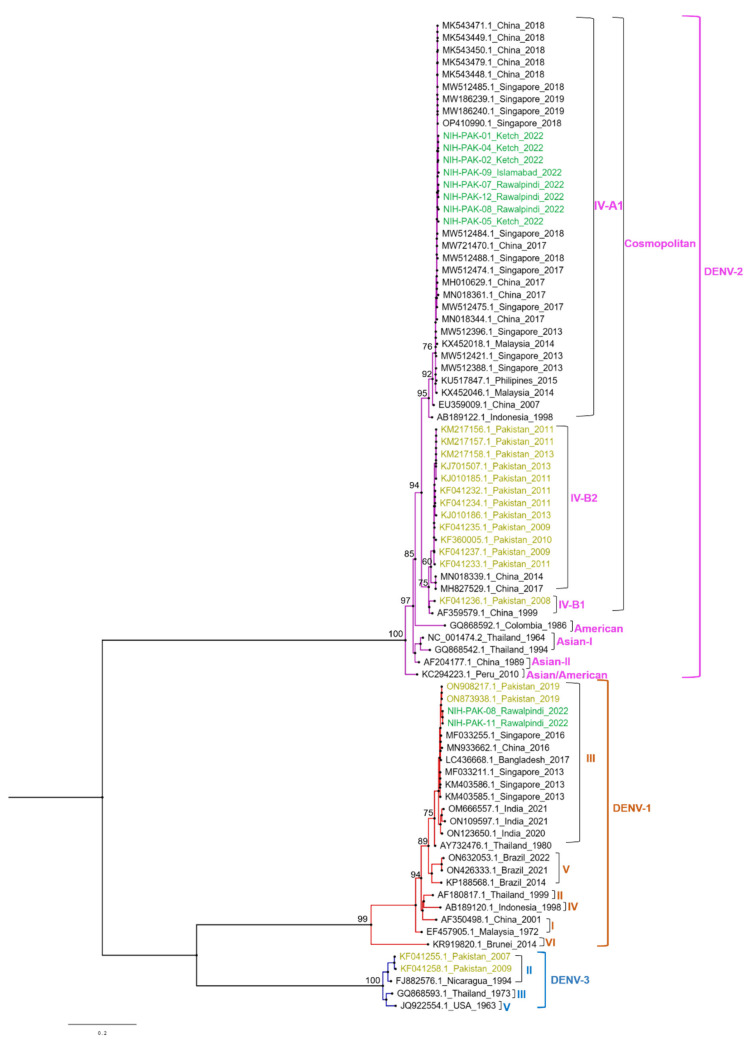
Maximum likelihood phylogenetic tree of dengue full genome sequences. The tree was generated through an IQ tree with 1000 bootstraps. The study isolates are highlighted in green, while the previously reported sequences of DENV from Pakistan are highlighted in yellow.

**Figure 4 vaccines-11-00163-f004:**
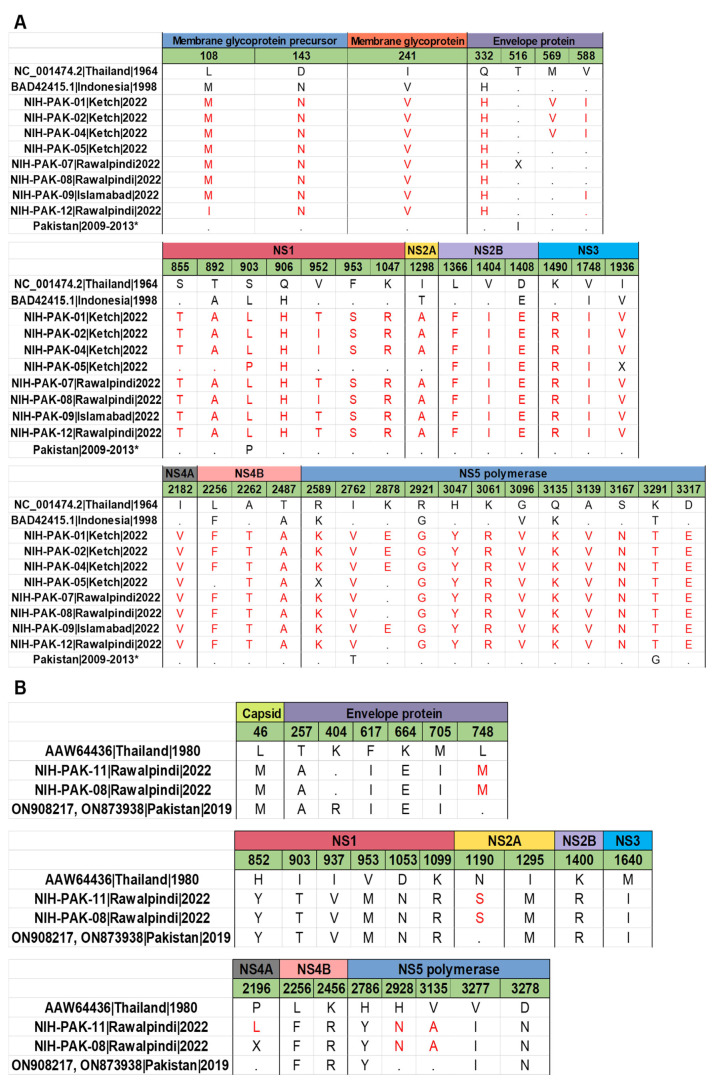
(**A**) Amino acid variation in the DENV2 study isolates in comparison with the reference strain and previously reported sequences from Pakistan. (**B**) Amino acid variation in the DENV1 study isolates in comparison with the reference strain and previously reported sequences from Pakistan. The amino acid differences are highlighted in red. X represents the missing amino acid at that position and dot (.) represents the same amino acid as present in the reference genome. * (AIU39216.1, AIU39217.1, AIU39218.1, AIE17400.1, AHM25909.1, AHC72404.1, AHC72406.1, AHC72407.1, AHC72405.1, AHM25910.1).

**Table 1 vaccines-11-00163-t001:** Demographic characteristics of Dengue patients.

Total No. of Cases (NS1+)	PCR Positive	PCR Negative
343	293 (85%)	50 (15%)
**Gender**	**No. (%)**	**No. (%)**
Male (213)	184 (63%)	29 (58%)
Female (130)	109 (37%)	21 (42%)
**Age groups (Years)**		
0–10	4 (1.3%)	1 (2%)
11–20	62 (21%)	8 (16%)
21–30	94 (32%)	17 (34%)
31–40	54 (18%)	13 (26%)
41–50	53 (18%)	8 (16%)
>50	26 (9%)	3 (6%)
	**Clinical Symptoms**	
	**Total available data = 223 patients**	
Fever	218 (98%)	
Myalgia	210 (94%)	
Backache	156 (70%)	
	**Hematological Markers**	
	**Total available data = 144 patients**	
Platelet Count	≤150,000 = 129 (89%) ≥150,000 = 15 (11%)	
Hematocrit Level	<40 = 96 (67%) 41–45 = 32 (22%) >45 = 15 (10%)	

NS1: non-structural protein-1; PCR: Polymerase Chain Reaction.

## Data Availability

All the sequences generated in the current study are submitted to GenBank-NCBI (https://www.ncbi.nlm.nih.gov/) under the accession numbers OP811977-OP811980, OP811982-OP811984, and OP898557-OP898559.

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
