# Peer review of "Genomic Characterization of Dengue Virus Outbreak in 2022 from Pakistan"

_vaccines, 2023, doi:10.3390/vaccines11010163_

Round 1
Reviewer 1 Report
The authors in the paper have investigated the diversity of circulating dengue virus responsible for an outbreak in Pakistan from August-October 2022. They have used real time RT-PCR CDC protocol for serotyping dengue virus. Authors further did genomic analysis on a few representative samples from DENV-2 and DENV-1 using Illumina next generation sequencing.
Comments:
1. In Page 1 line 44, there should be a comma between abdominal discomfort and vomiting.
2. Authors should mention all the results up to least one decimal point.
3. Authors should provide the information of ethical approval and patient consent in the method section.
4. In the method section the authors should use the term sample collection instead of data collection
5. Authors should classify the clinical characteristic of patients based on dengue virus serotypes.
6. Author should provide the NCBI accession numbers of sequences used in the present study.
7. In figure 3, figure legend is incomplete. What does the previously reported sequences of DENV from Pakistan are highlighted with?
8. The authors should include more samples for sequencing or should do Sanger sequencing for E and NS gene of dengue virus on significant number of samples.
9. Authors should discuss about the V952I along with V952T in Page 6 line 173.
10. Typo error in Page 8 line 259.
11. English needs to be improved
Reviewer 2 Report
This study characterizes the phylogenetic relationships of several newly sequenced dengue whole genomes. The genomes were sequenced from serum samples of NS1-confirmed dengue patients in 2022.
Introduction
Line 28 - "...one of the most important arboviruses...". Important for what?
Line 33 - "65% amino acid similarity" in which proteins?
Line 35 - It needs to be specified that DENV-5 does not currently circulate in humans.
Line 36 - The E gene has not been defined.
Line 39 - 42 - What do you mean by a varied immune response and how is this related to severe forms of dengue? What about the increased potential for severe disease with a second infection by a different serotype? The situation is more complex than how it is stated.
Most infections are asymptomatic and only a small percentage of symptomatic cases progress to DHF and DSS.
Line 49 - 3 million cases in 10 months is a lot less than the 100 million cases mentioned in the previous sentence. When did the period start?
Line 54 - How many cases equate to drastically increased?
Line 56 - Does this mean the 2021 outbreak continued into 2022?
Line 58 - This sentence is confusing. Are you talking about co-circulation of more than one serotype or all four at once? How does the low and high percentage of co-circulating serotype promote co-infection? Is there a certain percentage of a second serotype that increases the risk of co-infection?
Line 64 - How does surveillance prevent future outbreaks? It would seem that measures to prevent being bitten by mosquitoes would be more effective than knowing which serotypes are circulating. Additional evidence should be provided.
Methods
Line 72 - What other information was available for the samples? Age, sex, etc?
Line 77-80 - The RT-PCR information is unclear. Was serotyping done using a multiplex assay or a single-plex assay?
Line 83+ - Was there any PCR amplification of the virus done before library prep? Or was total RNA used as input for the library prep?
Line 100+ - These methods seem to be out of order. Wouldn't adapters be removed first, the reads trimmed and filtered for quality and then mapped to the reference?
Line 109 - What was the criteria for inclusion of sequences from GenBank?
Line 112 - How was the substitution model chosen?
Line 117 - Is the NS1 assay a serological test?
Line 129 - This statement does not reflect the figure below. DENV-2 was the dominant serotype until the week of 22-10-2022. DENV-1 did not reach greater than 50% until the last sampling week.
Figure 1 - Add how many samples are reflected in each column. What does w.r.t mean in the caption? Make the labels on the Y-axis black so they are easier to read.
Figure 2 - Labels on the circles are almost unreadable. Why not make pie charts so the labels can be larger?
Line 139 - By saying n=8 it suggests that there were only 8 DENV2 sequences in the analysis. This is obviously not the case. Did the sequencing result in 8 new whole genomes?
Line 141 - There is only a clade labeled IVA1 in figure 3.
Line 145 - Sequencing of these earlier samples should be in the methods and the details of the samples included unless the samples were sequenced prior to this study.
Line 150 - Please include the percent nucleotide and amino acid identities.
Line 155 - Meta-genome approach needs to be described in the methods. What were the criteria for including a sample in this approach? How many samples were included in the meta-genome approach? Was this the only one that successfully assembled two whole genomes?
What does the date refer to in the genome name? It is confusing for the reader when a sample collected in April has a genome date that is 6 months later.
Figure 3 - The figure needs to be larger. Why are there no bootstrap values? Clade names need a larger font. The caption needs to identify what the yellow indicate.
Line 165 - Please use the strain name not the GenBank accession number. What is the genotype of this strain?
Line 170 - What is meant by major amino acid substitutions?
Line 175 - They are significantly different because they are from different genotypes. This needs to be included because the statement makes it sound like a large number of mutations have occurred between 2013 and now.
Table 2 and 3 - The * should be a footnote to the table not in the title. If a consensus sequence was used to represent the previous sequences it needs to be mentioned. What does x mean? and what does the dot mean? These are not proper tables. They could be combined into a single figure with DENV2 (A) and DENV1 (B). Then the genomes would fit instead of being broken as shown.
Discussion
Line 197 - Infection or circulation?
Line 204 - What makes a significant outbreak? Number of cases, geographic distribution of reported cases?
Line 205 - This sentence should be rewritten to be clearer. What does n=3204 refer to?
Line 206 - National Institute of Health of Pakistan?
Line 208 - What is KP?
Line 209 - What is AJK?
Line 211-212 - Need to describe the connection between flooding and increase in dengue cases. Was it due to increase in mosquito populations? People being displaced from their homes and exposed to more mosquito bites? Both?
Line 215 - The serotypes are of symptomatic infections not prevalence in circulation. DENV2 is not the dominant cause of the sampled infections in the last 2 weeks of Oct according to Figure 1.
Line 217+ - Are you referring to serotypes of symptomatic infections or serotypes identified in mosquito collections? It is unclear. The source of the serotype data has different implications.
Line 225 - How does serotype analysis play a role in management of outbreaks?
Line 231 - This sentence is confusing. Was DENV3 the predominant serotype causing infections from 1990s to 2016 in Pakistan or just in Peshawar?
Line 233 - What is meant by a wide range of periods?
Line 240 - The phrase after IVA does not make sense.
Line 246-248 - This sentence is unclear. Was the 2017 DENV-2 the ancestor to clades IVA and IVB?
Line 250 -How was the outbreak in Peshawar dominated by Far East Asian countries?
Line 251 - The newly sequenced DENV2 genomes would obviously be different from sequences in another genotype. This does not equate to in situ evolution but more likely the result of an invasive strain that replaced the former genotype.
Line 260 - What regions of the genome were under different evolutionary pressures?
Line 274 - How did you decide what was a significant difference and what was not?
Line 276 - What references were the coinfected genomes compared to?
Line 285+ - Not all mutations act equally in different genotypes. The genotype with the T516I mutation is not the same as the currently sequenced genotypes. Sites under positive selection for 2008-2013 strains of one genotype should not be expected to be under the same selection for strains of a different genotype.
Line 293+ This observation is very confusing. Is the difference between DENV2 genomes from Pakistan or from 19 other countries? Are they in the same genotypes or different one?
Line 313 - Multiple serotypes are shown but there is only one genotype identified per serotype in the current study.
